# Macroglossia, Motor Neuron Pathology, and Airway Malacia Contribute to Respiratory Insufficiency in Pompe Disease: A Commentary on Molecular Pathways and Respiratory Involvement in Lysosomal Storage Diseases

**DOI:** 10.3390/ijms20030751

**Published:** 2019-02-11

**Authors:** Angela L. McCall, Mai K. ElMallah

**Affiliations:** Division of Pulmonary Medicine, Department of Pediatric, Duke University School of Medicine, Durham, NC 27710, USA; angela.mccall440@duke.edu

## Commentary

The authors of the recently published, “Molecular Pathways and Respiratory Involvement in Lysosomal Storage Diseases”, provide an important review of the various mechanisms of lysosomal storage diseases (LSD) and how they culminate in similar clinical pathologies [1]. Of note, the authors are quite thorough in their descriptions of the role of the diaphragm and the intercostal weakness in respiratory deficiency in Pompe disease. However, other major contributors to respiratory weakness in Pompe disease were not mentioned in this manuscript. In particular, macroglossia, the respiratory motor neuron pathology, and the airway malacia all contribute to respiratory insufficiency in Pompe disease. These topics have been reviewed elsewhere but merit is mentioned here, in the context of this review [2,3].

The authors describe the contribution of macroglossia to respiratory impairment in MPS-I, however they do not acknowledge the same condition within Pompe disease patients. As a result of large and weak tongues, infants with Pompe disease have problems maintaining upper airway patency. They often present with macroglossia, glossoptossis and tongue weakness [4,5,6]. In addition, tongue weakness leads to oral stage dysphagia and results in a weak suck and feeding problems early in life. As a result, these patients often fail to thrive. Furthermore, tongue weakness results in pharyngeal dysphagia, leading to difficulty with saliva management causing pooling of secretions and drooling. The pooled secretions increase the risk of aspiration and aspiration pneumonia. In late-onset Pompe disease, tongue weakness occurs even in otherwise asymptomatic patients, and obstructive sleep apnea is prevalent [7,8]. A study of 19 late-onset Pompe disease patients reports mild to severe tongue weakness in each of these patients. Furthermore, magnetic resonance imaging (MRI) reveals the diffusion of fatty infiltrates within the tongue [7].

As the authors mentioned, skeletal muscle weakness, including the diaphragm, is a clinical hallmark of Pompe disease. Glycogen accumulation within the diaphragm and intercostal muscles significantly contributes to respiratory morbidity. However, glycogen accumulation within the respiratory motor neurons also contributes to respiratory insufficiency [2,9]. For example, significant glycogen accumulation has been reported in the ventral horn of the cervical spinal cord of a toddler with Pompe disease. The ventral region of the spinal cord houses the phrenic motor neurons that innervate the diaphragm. In fact, central nervous system pathology has been observed upon patient autopsy, specifically within the anterior cervical spinal cord, anterior horn neurons, and brain stem neurons [10,11,12,13]. Furthermore, in the most widely studied Pompe disease, mouse model motor neuron dysfunction due to glycogen-filled lysosomes has a significant role in the weakness observed in the tongue and diaphragm [2,9,14,15,16,17].

In 2015, Yang et al. was the first to report airway malacia (trachea-broncho-malacia) in a teenage female with Pompe disease. Despite receiving ERT for 8 years, she continued to experience declining pulmonary function. In an exploration to elucidate her deteriorating respiratory system, a bronchoscopy was performed. This revealed a decreased luminal airway diameter along her lower airways. The left bronchus was nearly occluded with 90% of the lumen collapsed, requiring a metal stent to be implanted [18]. Similar observations were noted in a teenage male receiving ERT who had a progressive obstructive airway and restrictive pulmonary disease. While undergoing anesthesia, the patient developed ventricular defibrillation leading to an emergency bronchoscopy. The bronchoscopy revealed a collapse of the lower trachea and bronchi, leading to an almost completely compressed left main bronchus [19]. The mechanism of tracheal and bronchial weakness was studied in the Pompe mouse model, where it was found that Pompe airways are hyporeponsive to bronchocontrictive agents, and reduced calcium signaling occurs in Pompe airway smooth muscle cells [19]. These functional findings are also likely the result of gross glycogen accumulation and enlarged lysosomes present in the smooth muscle layer of the trachea and bronchi [3,20].

In conclusion, we agree with the authors that glycogen accumulation in the diaphragm and intercostal muscles plays an integral role in respiratory dysfunction in Pompe disease. However, glycogen accumulation in the tongue, the respiratory motor neurons, and the airway smooth muscle are also important contributors to the Pompe disease respiratory morbidities and should not be overlooked.

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
