# Peer review of "Macroglossia, Motor Neuron Pathology, and Airway Malacia Contribute to Respiratory Insufficiency in Pompe Disease: A Commentary on Molecular Pathways and Respiratory Involvement in Lysosomal Storage Diseases"

_ijms, 2019, doi:10.3390/ijms20030751_

Round 1

Reviewer 1 Report

The authors present relevant information to complement the description of the mechanisms involved in respiratory deficiency linked to Lysosomal Storage disorders, which was recently published by Faverio et al. 

In particular, the information, which was contributed in this article, is related to Pompe Disease, while in Faverio's paper the described mechanisms are referred to different LSDs.

The commentary format is appropriate for publication of this information, since this is a short compendium to the previously published article and not a full review on respiratory impairment in Pompe Disease.

Since the present paper is based on the Faverio's article, this reference should be added in the reference list.

Author Response

Thank you for your support of this manuscript and for your suggestion. The reference has been added. 

Reviewer 2 Report

Presented here commentary in form of review, brings comprehensive information about macroglossia, respiratory motor neuron pathology, and airway malacia role in respiratory insufficiency in Pompe disease. Authors accomplished to fulfill information lacking in publication commented. I can highly evaluate this short rewiev. Few colloquial words, like "massive fatty" in 29-30 line, might be replaced.

Author Response

Thank you for support of this commentary and for your suggestion. "Massive fatty infiltrates" has been changed to “diffuse fatty infiltrates”.

Reviewer 3 Report

The authors brief report complements an earlier review and underscores the multifactorial basis of respiratory problems in Pompe disease, which may ultimately impact on quality of response to current therapeutic option.

Minor, line 37 after CNS insert the word involvement

Author Response

Thank you for your support of this brief report.

Reviewer 4 Report

The authors add the known components of respiratory involvement but not been emphasized in the original paper. The commentary is sound and clear. 

Author Response

(The authors gave the same response as above.)
